# JAM-Flow: Joint Audio-Motion Synthesis with Flow Matching

## Abstract

The intrinsic link between facial motion and speech is often overlooked in generative modeling, where talking head synthesis and text-to-speech (TTS) are typically addressed as separate tasks. This paper introduces JAM-Flow, a unified framework to simultaneously synthesize and condition on both facial motion and speech. Our approach leverages flow matching and a novel Multi-Modal Diffusion Transformer (MM-DiT) architecture, integrating specialized Motion-DiT and Audio-DiT modules. These are coupled via selective joint attention layers and incorporate key architectural choices, such as temporally aligned positional embeddings and localized joint attention masking, to enable effective cross-modal interaction while preserving modality-specific strengths. Trained with an inpainting-style objective, JAM-Flow supports a wide array of conditioning inputs—including text, reference audio, and reference motion—facilitating tasks such as synchronized talking head generation from text, audio-driven animation, and much more, within a single, coherent model. JAM-Flow significantly advances multi-modal generative modeling by providing a practical solution for holistic audio-visual synthesis.

## 1 Introduction

With the rapid advancement of generative models Goodfellow et al. (2014); Ho et al. (2020); Song et al. (2021); Liu et al. (2023); Lipman et al. (2023); Rombach et al. (2022), the synthesis of realistic human faces and voices has become increasingly sophisticated. Two major fields have emerged from this trend: talking head generation Guo et al. (2024b); Xie et al. (2024); Siarohin et al. (2019); Drobyshev et al. (2024); Wang et al. (2021), which animates static portrait images to mimic facial expressions, and text-to-speech (TTS) synthesis Prajwal et al. (2020); Zhou et al. (2020); Tian et al. (2024); Xu et al. (2024b); Lin et al. (2025), which converts text and a short voice reference into natural-sounding speech. While state-of-the-art methods in both domains, ranging from GAN-based models Guo et al. (2024b); Zhang et al. (2023) for fast inference to diffusion-based models Xie et al. (2024); Lin et al. (2025) and flow matching-based models Chen et al. (2024); Jiang et al. (2025) for higher fidelity, have made remarkable progress, these two problems have traditionally been treated as separate tasks.

Yet in natural human communication, facial motion and speech are deeply interwoven. Movements of the mouth, cheeks, and jaw are not merely visual artifacts but integral components of spoken language. Surprisingly, despite this intrinsic connection, no prior work has jointly addressed talking head generation and speech synthesis in a unified model. Existing talking head models typically treat audio as a unidirectional condition, while TTS systems remain blind to facial dynamics.

In this work, we introduce the first training framework that can simultaneously model, generate, and condition on both audio and facial motion modalities within a single flow matching-based framework. We integrate two specialized flow matching models: a Motion-DiT for generating implicit facial keypoint sequences, and an Audio-DiT for denoising mel-spectrograms from text and reference speech. These modules are coupled through selective joint attention layers, where only half the layers are fused, allowing effective cross-modal communication while preserving the benefits of modality-specific representations. Furthermore, to inject structural inductive biases into the model, we incorporate rotary positional embeddings (RoPE) Su et al. (2024) with task-specific engineering improvements and attention masking to restrict temporal receptive fields.

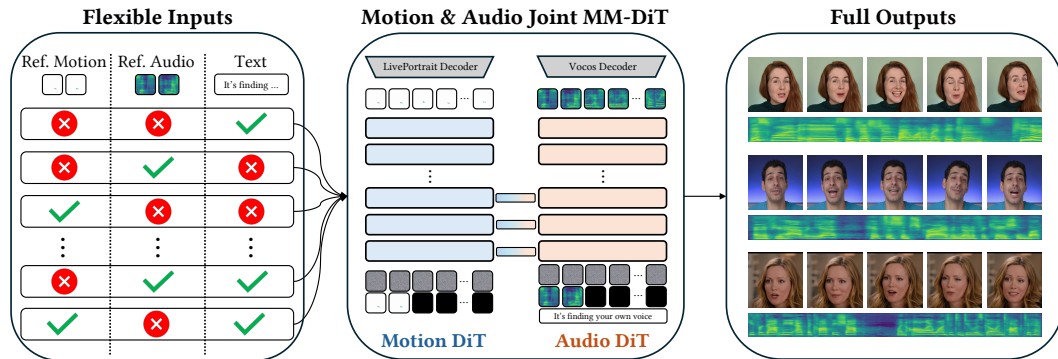

Figure 1: Overview of our JAM-Flow framework for flexible and joint generation of facial motion and speech. The model accepts diverse input combinations, including text, reference motion, and reference audio. These are processed by our novel Motion & Audio Joint MM-DiT, which enables synchronized synthesis of full audio-visual outputs supporting tasks like talking head generation from text, audio-driven animation, and cross-modal reconstruction (e.g., audio from motion).

One of our key experimental observations is that models should be initialized from well-trained components in their respective modalities before learning cross-modal relations. In our framework, we initialize the TTS branch with a pretrained F5-TTS model, while the Motion-DiT is first trained separately using the LivePortrait Guo et al. (2024b) framework to capture facial motion via compact keypoints. After this modality-specific pretraining, the two modules are jointly trained with shared attention layers and an inpainting-style supervision scheme. We propose that this joint inpainting-style supervision is particularly well-suited for optimizing pretrained unimodal models to learn robust cross-modal interactions, enabling flexible generation even under partially missing inputs.

Our joint inpainting-style supervision not only facilitates effective cross-modal learning but also enables versatile inference. By design, each modality can be conditioned on full, partial, or even absent inputs, allowing a single model to flexibly support diverse tasks and generation scenarios. As illustrated in Figure 1, our model supports a wide range of input-output configurations, including talking head generation, TTS, and cross-modal reconstruction, within a single coherent framework. To the best of our knowledge, this is the first attempt to employ joint inpainting-style supervision to achieve such flexible cross-modal synthesis, marking an important step toward practical and scalable multimodal generation.

Our contributions are summarized as follows:

1. We present the first joint training framework for talking head and TTS generation, which integrates both the model architecture and training strategy to enable mutual conditioning across modalities.

2. We demonstrate that our inpainting-style joint supervision effectively learns cross-modal relations, allowing robust and versatile generation across diverse tasks.

3. We further provide a practical architectural design for connecting pretrained unimodal models, including partial joint attention, RoPE integration, and attention masking, tailored for multi-modal flow matching.

## 2 RELATED WORK

### 2.1 FLOW MATCHING AND MULTIMODAL DIFFUSION TRANSFORMER

Flow matching Liu et al. (2023); Lipman et al. (2023); Esser et al. (2024) enhances generative modeling efficiency over score-based diffusion models Ho et al. (2020); Song et al. (2021). It learns a continuous transformation $Z_t$ between distributions via an Ordinary Differential Equation (ODE): $dZ_t/dt = v_\theta(Z_t, t)$, where $v_\theta$ is a learnable vector field. The objective is to match $v_\theta$ to a target velocity field $u_t(x)$. Conditional Flow Matching (CFM) Lipman et al. (2023) and Rectified Flow (RF) Liu et al. (2023) simplify this: for paired samples $x_0 \sim \pi_0$ and $x_1 \sim \pi_1$, they define an

intermediate point via linear interpolation, $x_t = (1 - t)x_0 + tx_1$, and set the target velocity $u_t(x_t)$ to be the difference $x_1 - x_0$. This results in the CFM loss:

$$\mathcal{L}_{\text{CFM}}(\theta) = \mathbb{E}_{t,x_0,x_1} \left[ \|v_\theta(x_t, t) - (x_1 - x_0)\|^2 \right]. \tag{1}$$

The *Multimodal Diffusion Transformer* (MM-DiT) Esser et al. (2024) builds on flow matching for joint multi-modal generation. It uses separate DiT Peebles & Xie (2023) branches per modality, fused via joint self-attention for cross-modal interaction, enabling scalable and expressive generation. Large-scale MM-DiTs like Stable Diffusion 3 Esser et al. (2024), Flux Labs (2024), and CogVideoX Yang et al. (2024) achieve SOTA in various tasks (e.g., text-to-image, video synthesis), but none have explored inpainting-style joint training to simultaneously synthesize two modalities.

## 2.2 TALKING HEAD HENERATION

Talking head generation synthesizes realistic facial animations from conditional signals, mainly through video-driven (facial reenactment) or audio-driven methods. Video-driven approaches Guo et al. (2024b); Xie et al. (2024); Siarohin et al. (2019) animate a source portrait using motion cues from a driving video. Early methods often used facial landmarks Siarohin et al. (2019) or 3D models Tewari et al. (2020), while recent works like LivePortrait Guo et al. (2024b) use implicit keypoints and X-portrait Xie et al. (2024) employs diffusion control. In contrast, audio-driven methods Prajwal et al. (2020); Zhou et al. (2020); Tian et al. (2024); Xu et al. (2024b) create lip movements synchronized with input audio. Pioneering work like Wav2Lip Prajwal et al. (2020) focused on lip-sync accuracy with GAN, while later methods (e.g., EMO Tian et al. (2024), Omni-Human Lin et al. (2025)) often use diffusion models for enhanced expressiveness.

Current methods typically model a uni-directional audio-to-visual flow. However, facial motion and speech are mutually influential in real conversations. Our hybrid approach addresses this by combining joint diffusion-based generation of keypoints and audio with efficient pixel decoding via LivePortrait, enabling bidirectional information exchange and flexible generation.

## 2.3 NEURAL TEXT-TO-SPEECH GENERATION

Neural text-to-speech (TTS) has progressed from attention-based sequence-to-sequence models Wang et al. (2017) to more advanced architectures. Early non-autoregressive systems Ren et al. (2019; 2020) enhanced efficiency via explicit duration modeling. A significant shift towards zero-shot voice cloning was pioneered by neural codec language models Wang et al. (2023), which autoregressively generate speech tokens from minimal reference audio, despite some inference challenges.

Diffusion-based methods have since emerged as a powerful paradigm, particularly for zero-shot voice cloning. These include approaches demonstrating high-quality speech via diffusion Tan et al. (2024); Ju et al. (2024); Shen et al. (2023); Le et al. (2023) and flow matching for efficient text-guided speech infilling Chen et al. (2024). Recent efforts focus on refining speech-text alignment and moving from explicit duration to more flexible frameworks Lee et al. (2024); Eskimez et al. (2024). F5-TTS Chen et al. (2024) advances this with flow matching and Diffusion Transformers (DiTs) for efficient non-autoregressive generation. While research continues to address alignment robustness, prosody, and efficiency Jiang et al. (2025), the simultaneous generation of speech and matching lip motion remains largely unaddressed.

## 2.4 AUTOMATED VIDEO DUBBING

Automated video dubbing synthesizes speech from text and video inputs, focusing on aligning generated speech with existing visual content, unlike video generation from audio/text. It extends text-to-speech (TTS) by conditioning on visual context, especially lip movements and facial expressions. The main challenge is ensuring temporal synchronization and reflecting visual expressiveness in the synthesized speech. Prior work Cong et al. (2023; 2024); Sung-Bin et al. (2025) has explored aligning visual cues, multi-scale style learning, and audio-visual fusion. Notably, our model, though not explicitly optimized for this task, shows an emergent capability for generating speech well-aligned with lip movements in given videos.

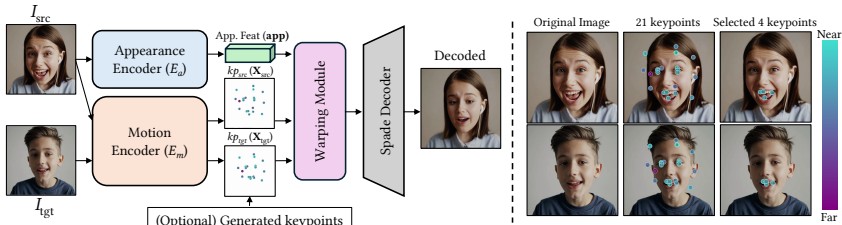

Figure 2: LivePortrait framework and mouth-related expression keypoint analysis. LivePortrait's motion encoder ($E_m$) infers parameters including a 3D expression deformation $\mathbf{e} \in \mathbb{R}^{21 \times 3}$ for 21 canonical keypoints. We find that deforming approximately four specific keypoints (highlighted) primarily dictates mouth articulation. Our Motion-DiT leverages this by generating only the deformation components ($\mathbf{e}^{\text{mouth}} \subset \mathbf{e}$) for these crucial mouth keypoints, enabling efficient lip-sync.

## 3 PRELIMINARIES

### 3.1 LIVEPORTRAIT FRAMEWORK

LivePortrait Guo et al. (2024b) enables image-to-video synthesis by disentangling structural and appearance features from a single input image. It employs an appearance encoder $E_a$ to extract a global appearance feature $\mathbf{app} \in \mathbb{R}^{C \times D' \times H' \times W'}$. In parallel, a motion encoder $E_m$ infers a set (21) of 3D implicit facial keypoints $\mathbf{X} \in \mathbb{R}^{21 \times 3}$ which is parametrized by canonical keypoints $\mathbf{x}_c \in \mathbb{R}^{21 \times 3}$, pose matrix $\mathbf{R} \in \mathbb{R}^{3 \times 3}$, expression deformation $\mathbf{e} \in \mathbb{R}^{21 \times 3}$, scale $\mathbf{s} \in \mathbb{R}$ and translation $\mathbf{t} \in \mathbb{R}^{21 \times 3}$. The final keypoints are then computed as: $\mathbf{X} = \mathbf{s} \cdot (\mathbf{x}_c \mathbf{R} + \mathbf{e}) + \mathbf{t}$. The warping module modifies $\mathbf{app}$ with esimated keypoint differences and the decoder projects this warped appearance feature $\mathbf{app}'$ into a target video frame.

Visualizing the 21 dimensions of the expression embedding $\mathbf{e}$ (Figure 2) reveals that approximately four specific dimensions consistently control the mouth region. Isolating these mouth-related components $\mathbf{e}^{\text{mouth}} \subset \mathbf{e}$ and freezing others modifies only the lip shape. This empirical finding suggests lip-sync generation can be simplified by modeling only this small subset of expression dimensions. We leverage this by training our motion generation module to predict only these mouth-related components, which are then combined with fixed identity- and pose-related features for rendering.

### 3.2 F5-TTS AND CONDITIONAL FLOW MATCHING

F5-TTS Chen et al. (2024) is a Conditional Flow Matching (CFM) based text-to-speech model. Inspired by inpainting-based approaches such as VoiceBox Le et al. (2023), F5-TTS treats speech generation as a mask-and-predict problem, where parts of the mel-spectrogram are randomly masked during training and then reconstructed conditioned on surrounding context and reference signals.

Formally, the model receives a masked audio segment $\tilde{\mathbf{a}}^{\text{masked}}$ and a conditioning vector $\mathbf{c}^{\text{text}}$ consisting of unmasked regions, reference audio features, and text embeddings. These are concatenated and passed through a flow-based network trained using the CFM objective described in Eq. 1.

This inpainting formulation has two key benefits. First, it enables robust training across diverse conditioning scenarios by randomly varying the masked regions. Second, it inherently models the relationship between unmasked context and the regions to be reconstructed, ensuring consistent speech generation. As a result, F5-TTS produces high-quality, voice-consistent speech for arbitrary text inputs and provides a strong foundation for our Audio-DiT module.

## 4 METHOD

### 4.1 OVERVIEW

Our goal is to simultaneously generate temporally aligned speech audio and facial motion from multimodal inputs (text, reference audio, or motion). To this end, we propose a dual-stream diffusion architecture composed of Audio-DiT and Motion-DiT, partially fused via joint attention blocks. Figure 3 illustrates the overall architecture with training and inference pipeline.

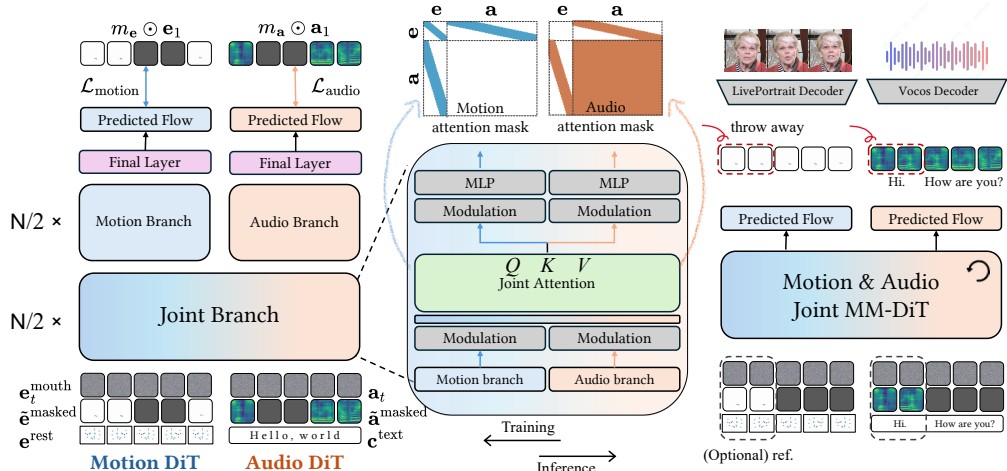

Figure 3: The training and inference pipeline of the JAM-Flow framework. Our joint MM-DiT comprises a Motion-DiT for facial expression keypoints ($\mathbf{e}^{\text{mouth}}$) and an Audio-DiT for mel-spectrograms ($\mathbf{a}$), coupled via joint attention. The model is trained with an inpainting-style flow matching objective on masked inputs and various conditions (text, reference audio/motion). At inference, it flexibly generates synchronized audio-visual outputs from partial inputs.

## 4.2 MOTION-DiT AND AUDIO-DiT DESIGN AND FLOW MATCHING

The Motion-DiT generates expression embeddings $\mathbf{e}^{\text{mouth}} \in \mathbb{R}^{T_{\text{frame}} \times 4 \times 3}$ that control lip motion, following the observation from Section 3.1 that 4 of the 21 expression dimensions are sufficient to model mouth dynamics. As shown in Figure 3, we provide two inputs to the motion stream: the target expression excluding mouth-related components, denoted $\mathbf{e}^{\text{rest}} \in \mathbb{R}^{T_{\text{frame}} \times 17 \times 3}$ and the audio-derived conditioning features $\mathbf{f}_{\text{audio}}$ from the Audio-DiT stream (if available). During training, the Motion-DiT denoises corrupted $\mathbf{e}^{\text{mouth}}$ vectors via a conditional flow-matching process:

$$\mathcal{L}_{\text{motion}} = \mathbb{E}_{\mathbf{e}_0^{\text{mouth}}, \mathbf{e}_1^{\text{mouth}}, t} \left[ \| v_\theta(\mathbf{e}_t^{\text{mouth}}, t; \mathbf{f}^{\text{audio}}, \tilde{\mathbf{e}}^{\text{masked}}, \mathbf{e}^{\text{rest}}) - (\mathbf{e}_1^{\text{mouth}} - \mathbf{e}_0^{\text{mouth}}) \|^2 \right] \quad (2)$$

where $\mathbf{e}_t$ denotes the noisy input at step $t$, and $v_\theta$ is the velocity predicting DiT.

The Audio-DiT generates mel-spectrograms $\mathbf{a}_{\text{mel}} \in \mathbb{R}^{T_{\text{mel}} \times d_{\text{mel}}}$ using the Conditional Flow Matching (CFM) objective. Following F5-TTS, we apply random inpainting masks to audio segments and condition on reference audio and text features. Additionally, we adapt the motion-derived conditioning features $\mathbf{f}^{\text{motion}}$ from the Motion-DiT stream (if available). As defined in Eq. equation 1, the model learns to reconstruct missing regions from context:

$$\mathcal{L}_{\text{audio}} = \mathbb{E}_{t, \mathbf{a}_0, \mathbf{a}_1} \left[ \| v_\theta(\mathbf{a}_t, t; \mathbf{f}^{\text{motion}}, \tilde{\mathbf{a}}^{\text{masked}}, \mathbf{c}^{\text{text}}) - (\mathbf{a}_1 - \mathbf{a}_0) \|^2 \right] \quad (3)$$

This enables the Audio-DiT to flexibly operate under partial or complete conditions and naturally generalize to reference audio-based speech generation.

The final flow-matching loss is defined as

$$\mathcal{L} = \mathcal{L}_{\text{audio}} + \mathcal{L}_{\text{motion}}. \quad (4)$$

During training, audio and motion streams are randomly masked: sometimes only motion is hidden, sometimes only audio, and at other times both are missing. This stochastic masking compels the model to attend jointly to the available context across modalities, naturally learning dependencies between unmasked and masked regions.

## 4.3 JOINT ATTENTION AND TEMPORAL FUSION

To enable cross-modal interactions, we introduce $N_{\text{joint}}$ layers of joint attention between the audio and motion streams, as illustrated in Figure 3. Tokens from both streams are fused via joint-attention blocks, while the rest of the transformer layers remain modality-specific.

Table 1: Comparison of talking head generation quality on HDTF Zhang et al. (2021) dataset.

| Method | FID ↓ | FVD ↓ | LSE-C ↑ | LSE-D ↓ |
|---|---|---|---|---|
| Ground Truth | - | - | 8.70 | 6.597 |
| SadTalker Zhang et al. (2023) | 22.340 | 203.860 | 7.885 | 7.545 |
| DreamTalk Ma et al. (2023) | 78.147 | 890.660 | 6.376 | 8.364 |
| AniPortrait Wei et al. (2024) | 26.561 | 234.666 | 4.015 | 10.548 |
| Hallo Xu et al. (2024a) | 20.545 | 173.497 | 7.750 | 7.659 |
| Hallo3 Cheng et al. (2024) | 20.359 | 160.838 | 7.252 | 8.106 |
| Ours-Stage1 | 18.372 | 194.27 | 7.138 | 7.947 |
| Ours (I2V) | 17.571 | 192.30 | 7.324 | 7.777 |
| Ours (V2V) | **11.633** | **25.07** | **8.086** | **7.181** |

To ensure temporal compatibility, we apply a simple alignment of rotary positional embeddings (RoPE). For position index $p \in [0, L-1]$ and frequency $\theta_d$, the rotation angle is defined as $\phi(p, d) = \frac{p}{L} L_{\text{ref}} \cdot \theta_d$, where $L$ is the sequence length of the current modality and $L_{\text{ref}} = \max(L_a, L_m)$. This scaling ensures that audio and motion tokens at the same timestep share comparable angular positions, despite having different sequence lengths. While this adjustment is straightforward, we found it to be indispensable: without such alignment, joint attention fails to converge, as queries and keys from different modalities interact at incompatible positional scales.

## 4.4 ATTENTION MASKING STRATEGY

A key novelty of our framework lies in the attention masking design for joint modeling. Unlike prior multimodal transformers (e.g., MM-DiT) that apply symmetric attention across streams, we introduce modality-specific asymmetric masks tailored to the functional roles of audio and motion.

For motion tokens, we impose a local self-attention window, reflecting the inductive bias that facial motion depends mainly on temporally adjacent frames. Motion-to-audio attention is restricted to the corresponding local window, while audio-to-audio attention is disabled during motion generation. Conversely, for audio tokens, we retain full global self-attention across the utterance, but restrict cross-modal attention to only motion tokens at the same timestamp, with motion-to-motion self-attention entirely masked out. (Figure 3, top)

This design is simple yet crucial: each modality attends in the way most natural to its generative process, while their cross-modal interactions remain tightly aligned. To our knowledge, no prior work has explored such asymmetric, modality-specific masking for joint attention. We found this strategy indispensable for stable training and synchronized outputs.

## 4.5 TRAINING PROCEDURE

Training is conducted in two stages:

**Stage 1:** We prepare pretrained models for each modality. For audio, we adopt F5-TTS as the pretrained TTS backbone. For motion, we train a model from scratch following a common practice in talking-head generation, where wav2vec2 features are concatenated to the motion input.

**Stage 2:** We then connect the two modalities with joint attention layers, applying RoPE alignment to ensure compatible temporal embeddings. Both branches are jointly trained with gradients flowing across modalities through the shared attention layers, enabling mutual refinement of audio and motion representations. At this stage, wav2vec2 features are no longer used: text conditioning is injected via the Audio-DiT, while the Motion-DiT consumes both $\exp_{\text{rest}}$ and intermediate audio features $\mathbf{f}^{\text{audio}}$. Further architectural and training details are provided in the supplementary material.

## 5 EXPERIMENTS

## 5.1 EXPERIMENTAL SETUP

We train our model on the CelebV-Dub Sung-Bin et al. (2025) dataset, filtered from CelebV-HQ Zhu et al. (2022) and CelebV-Text Yu et al. (2023). Training is conducted in two stages: the Motion-DiT

Table 2: Comparison of text-to-speech performance in LibriSpeech-PC test-clean Chen et al. (2024); Panayotov et al. (2015) benchmark. Methods are from prior work Du et al. (2024); Guo et al. (2024a); Eskimez et al. (2024); Chen et al. (2024); Jiang et al. (2025)

| Method | WER ↓ | SIM-o ↑ |
|---|---|---|
| Cosy Voice | 3.59% | 0.66 |
| FireRedTTS | 2.69% | 0.47 |
| E2 TTS | 2.95% | 0.69 |
| F5-TTS | 2.42% | 0.66 |
| MegaTTS 3 | **2.31%** | **0.70** |
| Ours | 4.91% | 0.64 |

Table 3: Comparison of automated video dubbing performance in CelebV-Dub Sung-Bin et al. (2025) dataset. Methods are from prior work Peng et al. (2024); Cong et al. (2023; 2024); Sung-Bin et al. (2025)

| Method | LSE-C ↑ | LSE-D ↓ | WER ↓ | spkSIM ↑ |
|---|---|---|---|---|
| Ground Truth | 6.73 | 7.44 | 4.15% | - |
| Zero-Shot TTS | 2.78 | 11.68 | 3.83% | 0.316 |
| HPMDubbing | **6.36** | **7.80** | 24.06% | 0.146 |
| StyleDubber | 3.78 | 10.40 | 9.48% | 0.264 |
| VoiceCraft-Dub | 6.05 | 8.33 | 7.01% | 0.333 |
| Ours | 3.43 | 10.56 | **6.39%** | **0.410** |

is first trained from scratch using keypoint representations, while the Audio-DiT is initialized from F5-TTS. After convergence, joint training is performed on paired audio-visual data. Evaluation is carried out on CelebV-Dub test splits and HDTF Zhang et al. (2021), using standard metrics including WER, SIM-o, LSE-C, LSE-D, spkSIM, FID, and FVD. *Note that no dataset currently provides aligned audio, video, and transcript annotations. CelebV-Dub includes pseudo-transcripts generated by Whisper, which can be inaccurate in many cases.*

## 5.2 QUANTITATIVE EVALUATION

Our model is the first to be explicitly trained for joint audio–motion generation, and thus there are no direct baselines for this unified setting. Nevertheless, by controlling which inputs remain unmasked at inference, the same model can flexibly perform multiple tasks, for example, audio-only, motion-only, or full audiovisual generation. To demonstrate its versatility, we evaluate across three representative tasks by comparing with models that are individually specialized for each, emphasizing that our approach does not rely on task-specific architectures yet still achieves competitive results.

**Talking head generation.** We evaluate talking head performance on HDTF using four key metrics: FID, FVD, LSE-C, and LSE-D. As shown in Table 1, our model performs competitively or better with SOTA methods such as SadTalker, AniPortrait, and Hallo3. Notably, our method is primarily designed with a video-to-video (V2V) setup in mind, utilizing a sequence of 17 non-mouth expression keypoints as a 'clue' (following Zhong et al. (2023)). This keypoint 'clue' is used in both V2V and image-to-video (I2V) configurations: V2V warps frames sequentially from a source video, whereas I2V consistently warps an initial source image.

**Text-to-speech generation.** TTS performance is evaluated using WER and SIM-o on LibriSpeech-PC test-clean Chen et al. (2024); Panayotov et al. (2015). As shown in Table 2, our scores are slightly worse than those of dedicated TTS systems. We emphasize that our model is not designed as a pure TTS model but as a unified audio–motion generator, so some degradation is expected. Importantly, the dataset used for training CelebV-Dub relies on Whisper-generated captions that contain many transcription errors (see supplementary for analysis), and on demixed audio obtained via Spleeter, both of which limit achievable TTS quality. In fact, the WER of our model aligns closely with the Whisper-base error rate (4.50%) reported on LibriSpeech-PC Chen et al. (2024) (Table 3 in the original paper), suggesting that dataset noise rather than model design is the main bottleneck.

**Automated video dubbing.** Thanks to the inpainting-based training paradigm, our model naturally extends to automated video dubbing, generating speech that is both semantically correct and temporally aligned with the speaker's lip movements. However, as discussed by Yaman et al. (2024); Muaz et al. (2023); Sung-Bin et al. (2025), we found that SyncNet-derived metrics Chung & Zisserman (2017) (LSE-C, LSE-D) often fail to operate reliably, showing instability under codec variations and, most critically, with TTS outputs from our model. We therefore caution against over-interpreting these scores and refer readers to the supplementary materials for qualitative examples that better

Table 4: Ablation study on the number of joint attention blocks, motion attention masking, and Audio-DiT finetuning.

| '# Joint Blocks | Attn. Mask | Train Audio-DiT | FID ↓ | LSE-C ↑ | LSE-D ↓ | WER ↓ | SIM-o ↑ |
|---|---|---|---|---|---|---|---|
| 22 (Full) | ✗ | ✗ | 5.759 | 4.81 | 8.40 | 6.88% | 0.62 |
| 11 (Half) | ✗ | ✗ | 5.735 | 4.64 | 9.07 | 7.25% | 0.62 |
| 11 (Half) | ✗ | ✓ | 5.748 | 6.44 | 7.99 | 7.93% | 0.61 |
| 11 (Half) | ✓ | ✗ | 5.747 | 5.76 | 8.22 | **6.76%** | 0.63 |
| 11 (Half) | ✓ | ✓ | **5.662** | **6.45** | **7.73** | 7.28% | **0.64** |

reflect performance. As seen in Table 3, our model achieves strong WER and the highest spkSIM among all methods, demonstrating its effectiveness for this task.

## 5.3 QUALITATIVE RESULTS

We provide a supplementary webpage with videos. On standard settings (talking-head comparison, TTS comparison, and automated dubbing), our method shows stronger lip–audio alignment and speaker consistency; please see the supplement for side-by-side examples and details.

**Our Exclusive use cases.** **Case 1 Text → Audio + Motion**: Without any reference audio, the model *jointly* generates random identities' speech and facial motion that remain tightly synchronized. This text-only joint generation is a primary target of our training. **Case 2 Text + Reference Audio →** **Audio + Motion**: Given target text and a short voice reference, the model co-generates speech in the reference voice and the matching lip motion. This is likewise a core *joint* audio–motion scenario we explicitly train for. **Case 3 Reference Motion + Target Text → Audio**: With the video fixed, the synthesized audio adapts its opening/closure timing to the observed lip motion, even when perfect articulation is impossible, indicating active attention to motion during speech generation. **Case 4 Reference Motion → Audio (no text)**: Supplying only motion still yields plausible, time-aligned speech, showing an implicit mapping from visual articulation to acoustics under partial conditioning.

These results highlight the controllability and robustness of our unified model under partially missing inputs; see the supplementary webpage for qualitative examples.

## 5.4 ABLATION STUDIES

To assess the impact of our design choices, we conduct controlled ablation experiments along three axes: (1) the degree of joint attention between audio and motion streams, (2) the presence of temporal attention masks in the Motion-DiT, and (3) the finetuning of the Audio-DiT during stage-2 training. Given the aforementioned instability of LSE-C and LSE-D with generated audio, and to better assess the commonly adopted audio-driven talking head setup, we calculate LSE-C and LSE-D in the ablation study using generated motion paired with ground truth (GT) audio. WER and SIM-o are computed using both generated motion and audio, while LSE-C and LSE-D use GT audio for stability.

**Joint Attention Configuration.** We compared a *Full Joint* configuration (all DiT layers share attention) with our *Half Joint* approach (only earlier layers fused). Although Table 4 shows numerically stronger scores for Full Joint, it was both unstable and computationally much heavier. In practice, we found that Half Joint provided a more reliable trade-off: comparable qualitative performance with significantly lower training cost. Our choice of Half Joint therefore reflects a balance between performance and efficiency, which proved most practical for large-scale experiments.

**Attention Masking (and RoPE).** Removing temporal attention masks (*No Masking*) leads to sharp drops in lip-sync quality. While the degradation may not always be dramatic in raw scores, subjectively the model often fails to achieve coherent joint training at all, producing temporally drifting or unsynchronized motion. In other words, masking is not just helpful but almost indispensable for stable learning. Similarly, RoPE alignment (Section 4.3) is absolutely critical: without it, joint training does not converge at all, which is why we do not report a corresponding ablation in the table.

**Audio-DiT Finetuning.** We also tested freezing the Audio-DiT during stage-2 training to preserve the strong F5-TTS prior. As shown in Table 2 and Table 4, this setting slightly improves WER but results in weaker synchronization. Our experiments revealed why: keeping Audio-DiT fixed prevents the system from learning a proper joint audio–motion distribution, leaving Motion-DiT to adapt alone. Allowing Audio-DiT to finetune, by contrast, enables both modalities to co-adapt through the shared attention layers, yielding more synchronized and coherent outputs.

Taken together, our experiments show that Half Joint attention, modality-specific masking with RoPE alignment, and Audio-DiT finetuning are essential not only for quantitative gains but also for stable and efficient joint training that truly learns a shared multimodal distribution.

## 6 DISCUSSION AND ETHICS

Our experiments show that partial joint attention and localized temporal masking are key to stable and coherent multimodal generation. While full joint attention yields slightly better scores, it often results in unstable training. Our hybrid design balances cross-modal fusion and modality-specific representations, contributing to both robustness and performance.

We also observed that the generated speech often reflects emotional cues from facial motion. For instance, smiling motions tend to produce brighter, higher-pitched voices, despite the absence of explicit emotion supervision. This suggests the model implicitly aligns emotion across modalities, opening possibilities for expressive and emotionally-aware generation.

A major strength of our model is its versatility. It supports diverse input configurations (e.g., audio-only, motion-only, text + portrait) within a single framework, unlike prior models limited to either talking head synthesis or TTS. This flexibility enables applications such as expressive dubbing, silent video revoicing, and adaptive avatar generation.

Beyond our current scope, we found that generating two modalities jointly is remarkably effective, and we believe this paradigm can naturally extend to other modality pairs (e.g., depth + video, audio + video). In particular, our results suggest that inpainting-style joint supervision provides a powerful mechanism for learning cross-modal relationships, and merits deeper exploration as a general framework for multimodal co-generation.

The JAM-Flow model also raises ethical considerations. While it provides benefits for accessibility, avatars, and creative tools, it may also be misused for deepfakes or amplify biases in the data. To address these risks, we plan to explore safeguards such as watermarking and continue reflecting on responsible deployment as the technology evolves.

## 7 CONCLUSION

We presented the first joint training framework for speech and facial motion generation, integrating model architecture and training strategy to enable mutual conditioning across modalities. Our unified flow-matching-based design combines modality-specific DiT modules with selectively applied joint attention, RoPE alignment, and attention masking, producing coherent multimodal synthesis without separate pipelines. Through extensive experiments, we demonstrated that inpainting-style joint supervision effectively learns cross-modal relations and supports flexible inference scenarios such as motion-to-audio, audio-to-motion, and full generation from text and portrait image alone, suggesting it as a powerful paradigm for multimodal co-generation. Finally, we provided a practical architectural pathway for connecting pretrained unimodal models, showing that partial joint attention, RoPE alignment, and localized masking together balance stability, performance, and efficiency, yielding competitive results with specialized state-of-the-art models.

While these findings are promising, current limitations stem largely from data and compute. CelebV-Dub Sung-Bin et al. (2025), for instance, contains Whisper-generated captions with transcription errors and demuxed audio with artifacts, while LivePortrait constrains motion modeling largely to facial regions. Nevertheless, we believe our results make a compelling case for unified multimodal training. With more curated datasets and stronger video diffusion backbones Yang et al. (2024); HaCohen et al. (2024); Wan et al. (2025), future work can further extend this framework, unlocking expressive dubbing, controllable avatars, and more natural human–computer interaction.

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

## A EXPERIMENTAL DETAILS

**Datasets**  We train on CelebV-Dub Sung-Bin et al. (2025), filtered from CelebV-HQ Zhu et al. (2022) and CelebV-Text Yu et al. (2023). Videos are up to 30 seconds in length (the average is much shorter). For evaluation, clips longer than 20 seconds are segmented into non-overlapping 20-second chunks. Unless otherwise noted, we evaluate on the CelebV-Dub test split and HDTF Zhang et al. (2021).

**Audio features.**  We extract 100-dimensional mel-spectrograms with FFT size 1024, hop length 256, window size 1024, and target sampling rate 24 kHz.

**Motion representation.**  Facial motion is modeled at 25 FPS using preprocessed keypoint trajectories. We also define *rest_motion* as the set of facial keypoints excluding the mouth region.

**Known data limitations.**  CelebV-Dub contains pseudo-captions generated by Whisper and demixed audio obtained via Spleeter. We observed non-trivial transcript errors and occasional artifacts in the demuxed audio. A quantitative breakdown and examples are provided in the supplement.

**Backbones.**  Audio-DiT and Motion-DiT share the same transformer backbone: hidden dimension 1024, depth 22, 16 attention heads (dim_head 64), dropout 0.1.

**Joint attention layers.**  We insert $N_{\text{joint}}$ joint-attention blocks to couple the two streams. Unless otherwise specified, we adopt a *Half Joint* configuration that fuses the early portion of the network (first half of the 22 layers) and keeps later layers modality-specific. This balances cross-modal fusion and modality-specialized capacity with substantially lower compute than *Full Joint*.

**Stage 1 (unimodal preparation) details.**  Audio: initialize Audio-DiT from pretrained F5-TTS checkpoints. Motion: train Motion-DiT from scratch using keypoint inputs. During this stage we follow the talking-head practice of concatenating wav2vec2 features to motion inputs; these features are not used in Stage 2.

**Stage 2 (joint training) details.**  Connect the two streams via joint-attention layers and apply RoPE alignment. Text conditioning is injected through Audio-DiT. Motion-DiT consumes rest_motion ($\exp_{\text{rest}}$) and intermediate audio features $\mathbf{f}^{\text{audio}}$. Gradients flow across modalities within joint blocks, enabling mutual refinement.

**Conditioning dropout.**  To encourage robustness and enable conditional sampling, we apply modality-specific dropout during training:

```
cond_audio_drop_prob=0.1,  cond_motion_drop_prob=0.1,
text_drop_prob=0.2,  rest_motion_drop_prob=0.8.
```

These drops randomly mask conditioning signals, matching the inpainting-style supervision used throughout training.

**Classifier-Free Guidance (CFG)**  We support controllability via CFG by randomly dropping joint conditioning during training. Concretely, we apply a 10% dropout on joint-attention conditioning to create paired conditional and unconditional predictions. At inference, a guidance scale $\gamma$ blends the two predictions to adjust the strength of multimodal conditioning. We use $\gamma = 2$ for all reported results, which we found to balance synchronization and realism. Larger values can exaggerate articulation; smaller values reduce expressiveness. The model remains robust even without CFG, indicating strong learned cross-modal representations.

**Optimization and Compute**  All experiments run on 4 NVIDIA RTX 6000 Ada GPUs. Training a single model variant across both stages takes about one day. We use 32 NFEs for all samplings, consistent with F5-TTS.

## B  Qualitative Comparisons, User Study, and Discussions

### B.1  Qualitative Analysis

We provide extensive qualitative comparisons across five categories to complement the quantitative results in the main paper. The supplementary HTML file contains:

- **Talking Head Generation**: 14 test samples from HDTF Zhang et al. (2021) dataset comparing our method (both I2V and V2V variants) against SadTalker Zhang et al. (2023), AniPortrait Wei et al. (2024), Hallo Xu et al. (2024a), and Hallo3 Cheng et al. (2024). Our V2V variant demonstrates superior lip-sync accuracy and more natural facial dynamics. Our approach achieves noticeably better lip synchronization with the given audio. Importantly, because our method uses keypoints beyond the mouth region as conditioning clues, it preserves the source motion and gestures more faithfully, leading to outputs that are both lip-synced and motion-consistent.

- **Text-to-Speech**: 10 test samples from LibriSpeech-PC test-clean Panayotov et al. (2015); Chen et al. (2024) comparing our method against F5-TTS baseline. While standalone performance trails specialized TTS models due to pseudo-caption training, our audio quality excels as an engine for diverse creative applications, enabling audio-visual generation and cross-modal conditioning beyond what pure TTS models can achieve. Although our model reaches a WER of 4.9%, it preserves speaker identity effectively and produces speech that listeners judge as natural and consistent. We also report results from Ours$^\dagger$, where the Audio-DiT is completely frozen, essentially replicating F5-TTS performance. This comparison highlights that our unified model, despite not being designed as a TTS system, remains competitive in terms of intelligibility and voice similarity.

- **Automated Video Dubbing**: 15 test samples from CelebV-Dub Sung-Bin et al. (2025) comparing against HPMDubbing Cong et al. (2023), StyleDubber Cong et al. (2024), and VoiceCraft-Dub Sung-Bin et al. (2025). Our method achieves synchronized audio-visual generation without explicit optimization for this task. Quantitative metrics such as LSE-C and LSE-D are unreliable in this setting (see main paper discussion), so qualitative inspection is particularly important. We selected random test videos; while our model occasionally produces errors in matching prompts, comparison methods also fail under similar conditions. Overall, qualitative results demonstrate clear advantages in temporal alignment and speaker similarity.

- **Exclusive Use Cases**: We showcase some unique generation capabilities that emerge from our joint audio-motion modeling framework. These include: (1) text-to-multimodal synthesis generating both audio and motion from text alone, (2) voice-preserving multimodal generation using reference audio for speaker identity, (3) motion-constrained audio synthesis where frozen motion guides audio generation with different semantic content, and (4) motion-to-audio generation without any text cues, demonstrating the model's ability to infer plausible speech from visual patterns alone. These diverse conditioning scenarios highlight the flexibility and cross-modal understanding of our unified approach.
  - Case 1: Text → Audio + Motion. With only text input, the model *jointly* generates both speech and synchronized lip motion, even without reference audio. The generated voices are drawn from random speaker identities, yet remain coherent with the motion. This setting directly validates the primary purpose of our training: simultaneous audio–motion generation. • Case 2: Text + Reference Audio → Audio + Motion. Similar to Case 1, but now conditioned on a short reference audio clip. The model produces speech in the target voice while generating matching lip motion. This again reflects the central joint audio–motion training objective, showing that the system can both preserve voice identity and synchronize motion to new text. • Case 3: Reference Motion + Target Text → Audio. Here, we fix the video and change the text. Perfect lip–audio alignment is impossible, since motion cannot be altered. Nevertheless, the generated audio aligns its timing (mouth opening/closure) with the visible motion, revealing that motion cues are attended to during speech generation. This case provides experimental evidence that unmasked motion features influence audio generation. • Case 4: Reference Motion → Audio (without text). In this extreme setting, only motion is provided while text input is dropped. The model still produces plausible, time-aligned speech, showing that motion alone can guide audio synthesis. This demonstrates

that the model has implicitly learned a mapping between visual articulation and acoustic patterns, beyond text-based conditioning.

- **Failure Cases**: Analysis of current limitations, including (1) synchronization failures when input modalities exhibit significant length mismatches—while minor discrepancies are handled through natural interjections, severe misalignments break lip-sync coherence, and (2) degraded performance when LivePortrait base model fails to detect keypoints on non-realistic inputs such as flat cartoons or highly stylized artwork.

## B.2 LIBRISPEECH-PC PERFORMANCE AND DATASET CONSIDERATIONS

Our model differs fundamentally from pure TTS systems in that it jointly optimizes for both audio and motion generation. Consequently, the observed WER gap (4.91% vs. 2.42%) reflects a multi-objective optimization trade-off. Ours$^{\dagger}$), a frozen Audio-DiT achieves 3.38% WER, suggesting that part of the gap arises from dataset-related factors such as Whisper-generated captions and background music removal.

A key challenge is the absence of publicly available datasets containing aligned triplets of video, audio, and transcript. CelebV-Dub Sung-Bin et al. (2025) was therefore adopted as the only available proxy, constructed from CelebV-HQ Zhu et al. (2022) and CelebV-Text Yu et al. (2023) via background music removal and Whisper-based transcription. While this dataset enables joint training, it inevitably introduces transcription errors and audio artifacts.

**Transcript quality.** We manually inspected all 213 test videos of CelebV-Dub, comparing audio against Whisper transcripts. Approximately 20% of samples contained at least one incorrect word (45/213), and 30% omitted the final word (72/213). The latter likely stems from Whisper's difficulty in timestamping the last token. Hallucinations were also observed, e.g., "They won't" transcribed as "They won't stop talking about it."

**Impact of background music removal.** We evaluated whether the demuxing step itself degrades transcription. Applying the same background-music removal pipeline to clean F5-TTS outputs increased WER from 2.43% to 2.63%, suggesting that preprocessing artifacts may indeed contribute to the higher WER observed in CelebV-Dub.

**Video modality noise.** CelebV-Dub originates from YouTube videos recorded in uncontrolled environments, often with distant microphones, background noise, or sound effects. These factors introduce additional variability not present in curated audio-only corpora.

Taken together, these analyses suggest that the higher WER of our model is not solely a reflection of model quality, but also of dataset limitations. Our framework is not designed as a specialized TTS system; rather, it pioneers a unified formulation of joint speech–motion generation. Within this context, the performance observed on LibriSpeech-PC is consistent with both the multi-objective nature of our model and the imperfections of current pseudo triplet datasets. Future work may benefit from more carefully curated datasets and advanced preprocessing pipelines, but our focus here is to demonstrate the feasibility and effectiveness of a unified architecture for multimodal co-generation.

## B.3 USER STUDY

We conducted a user study with 26 participants to evaluate perceptual quality across two tasks:

(1) **Audio-Conditioned Talking Head Generation (HDTF):** Participants ranked six methods from best to worst for each sample. As shown in Figure A1, our V2V variant achieved the best average rank of 1.29, followed by our I2V variant (2.28), significantly outperforming SadTalker (5.04), AniPortrait (5.51), Hallo (3.02), and Hallo3 (3.85).

(2) **Automated Video Dubbing (CelebV-Dub):** Participants selected the best model among four methods for each sample. Figure A2 shows that our method received 62.6% of votes, demonstrating a strong preference over VoiceCraft-Dub (37.4%), while HPMDubbing and StyleDubber received no votes.

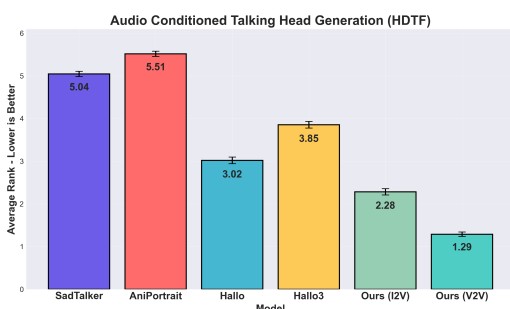
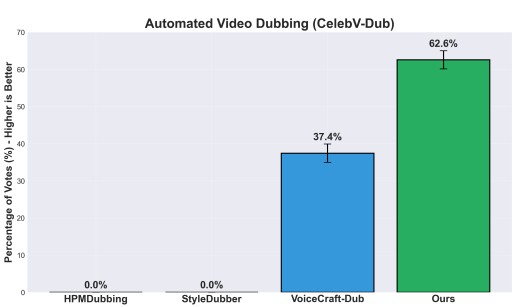

Figure A1: Average ranking results for audio-conditioned talking head generation on HDTF dataset. Participants ranked six methods from best (1) to worst (6) based on overall quality, including lip-sync accuracy, motion naturalness, and visual fidelity. Lower rank indicates better performance.

Figure A2: User preference results for automated video dubbing on CelebV-Dub dataset. Participants selected the best synchronized audio-visual output among four competing methods for each sample. Values indicate the percentage of times each method was chosen as best.

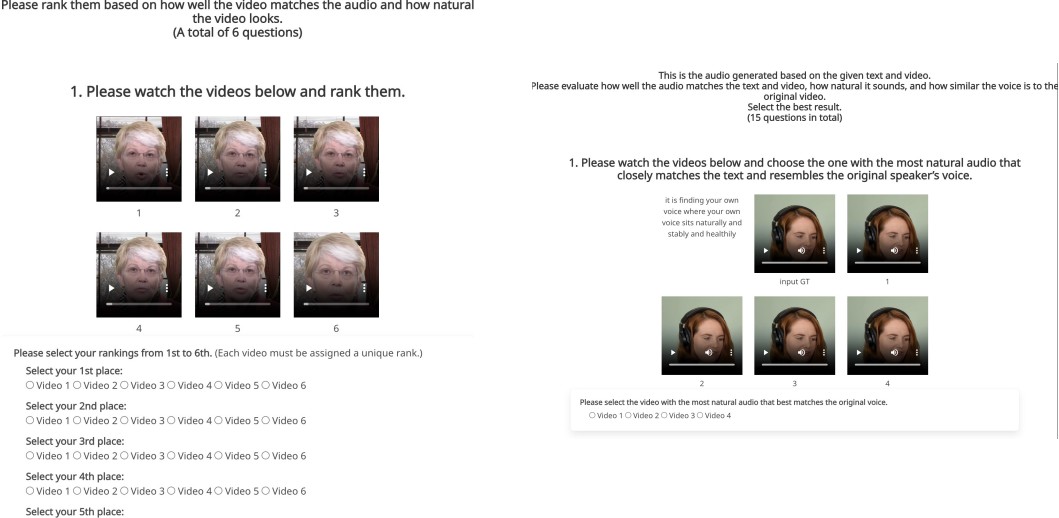

Figure A3: Survey Examples. The left shows an example from the Audio-Conditioned Talking Head Generation (HDTF) survey, and the right shows an example from the Automated Video Dubbing (CelebV-Dub) survey.

### B.4 ADDITIONAL DISCUSSIONS

Beyond the quantitative metrics and user study results, several interesting patterns emerged during our extensive experiments. In talking head generation scenarios, our method leverages the efficient and compact LivePortrait implicit keypoint warping approach, which not only accelerates inference but also effectively preserves identity and fine-grained facial details. Throughout our evaluations, we observed that while diffusion-based methods such as Hallo and Hallo3 can handle complex scenes, they tend to exhibit temporal flickering artifacts that become particularly noticeable in longer sequences.

For audio generation in dubbing scenarios, our approach benefits from the rich knowledge embedded in pre-trained TTS models, which contributes to its superior performance compared to other dubbing methods. The generated audio consistently demonstrates remarkable alignment with lip motions, capturing subtle nuances including natural pauses during speech breaks and maintaining coherence between facial expressions and vocal tone. This emergent coherence was not explicitly supervised

during training, suggesting that our joint modeling approach captures deeper cross-modal relationships beyond simple temporal synchronization.

An intriguing observation emerges when examining the model's behavior during simultaneous audio and motion generation. The system exhibits an asymmetric adaptation strategy, primarily modifying the audio to match the given motion rather than significantly altering the visual content. In practice, this manifests as minimal adjustments to lip movements while the audio undergoes more substantial modifications to achieve perfect synchronization. This behavior likely reflects the inherent challenge of modifying visual motion compared to audio synthesis and represents a practical solution that prioritizes visual consistency while ensuring accurate lip-sync.

These qualitative advantages, combined with strong user preference results, demonstrate that joint audio-motion modeling within a unified architecture produces more natural and synchronized talking head videos compared to existing cascade or single-modal approaches. We encourage readers to explore the "Exclusive Use Cases" and "Failure Cases" sections for deeper insights into our model's behavior and limitations.

### B.5 INFERENCE SPEED

Our method's compact representation space enables remarkably fast inference. We used default settings for all baseline methods and 32 NFE (Number of Function Evaluations) for our method as specified in the main paper. On a single RTX A6000 GPU, generating a 20-second lip-synced HDTF sample takes:

- *Our method*: 45 seconds / $\sim$500M Parameters (Joint audio+motion)
- *SadTalker*: 2.5 minutes (with GFPGAN) / $\sim$200M Parameters
- *AniPortrait*: 7 minutes / $\sim$1B Parameters
- *Hallo*: 23 minutes / 1$\sim$3B Parameters
- *Hallo3*: 30 minutes (on H100 GPU) / 1$\sim$3B Parameters

Note that Hallo3 requires GPUs with >48GB memory and was tested on an H100 GPU (known to be several times faster than A6000). Our training utilized 4 RTX 6000 Ada GPUs, while inference benchmarks were conducted on a single RTX A6000, except for Hallo3, which required external cloud infrastructure with an H100 GPU.

These results demonstrate a notable speedup over existing methods, making our approach practical for real-world applications while maintaining superior quality, as validated by our user studies.

### C DETAILS ON USED DATASETS AND MODELS

Our models, including the first-stage Motion-DiT, were trained exclusively on the CelebV-Dub Sung-Bin et al. (2025) dataset, which builds upon the CelebV-HQ Zhu et al. (2022) and CelebV-Text Yu et al. (2023) datasets. Both CelebV-Text and CelebV-HQ were sourced from the internet by their respective authors and are available solely for non-commercial research purposes. While CelebV-Dub does not explicitly specify licensing restrictions, we adhere to the same non-commercial usage constraints as its underlying datasets.

The only pre-trained model utilized in our work, F5-TTS Chen et al. (2024), is licensed under CC-BY-NC. Given these licensing constraints from both our training data and pre-trained models, our model, to be released in the future, will similarly be available under CC-BY-NC license for non-commercial research purposes only. This licensing may be subject to revision should the terms of the underlying datasets or models change.

### D DETAILS ON JOINT ATTENTION IMPLEMENTATION

We provide the implementations in Algorithm 1 and Algorithm 2. To support modality-specific masking and enable configurations where audio joint attention can be disabled (e.g., for classifier-free guidance), we implemented the joint attention mechanism as shown in Algorithm 2. During training,

all conditioning inputs (text, audio, and motion) are randomly dropped following the standard CFM training strategy. Rotary positional embeddings (RoPE) are scaled according to the length of the audio to ensure temporal alignment across modalities. The code will be publicly released to support reproducibility and further research.

## E    ABOUT PAPER

**The Use of Large Language Models (LLMs)**    This paper only employed LLMs for polishing the text, not for generating content.

**Ethics**    Ethical considerations must be taken into account when developing and deploying such models.

**Reproducibility Statement**    The code will be released at a later stage to ensure reproducibility.

**Algorithm 1: JointDiTBlock** — single block of a diffusion transformer operating on two sequences with optional joint cross-attention. Element-wise multiplication is denoted by $\odot$.

**Input:** $x_1, t_1$ — hidden state & timestep for branch 1
$x_2, t_2$ — hidden state & timestep for branch 2
$B_1 = (\text{attn\_norm}_1, \text{attn}_1, \text{ff\_norm}_1, \text{ff}_1)$,
$B_2 = (\text{attn\_norm}_2, \text{attn}_2, \text{ff\_norm}_2, \text{ff}_2)$
$\text{mask}_1, \text{mask}_2$ — optional local-window masks
$\text{rope}_1, \text{rope}_2$ — rotary position embeddings
$\alpha_1, \alpha_2 \in \{0, 1\}$ — flags: use *joint* attention for each branch
**Output:** $x_1', x_2'$ — updated hidden states

/* 1. Normalization & gating coefficients */
$(\hat{x}_1, g_1^{\text{msa}}, s_1^{\text{ff}}, e_1^{\text{ff}}, g_1^{\text{ff}}) \leftarrow B_1.\text{attn\_norm}(x_1, t_1)$
$(\hat{x}_2, g_2^{\text{msa}}, s_2^{\text{ff}}, e_2^{\text{ff}}, g_2^{\text{ff}}) \leftarrow B_2.\text{attn\_norm}(x_2, t_2)$

/* 2. Joint or independent multi-head attention */
$(a_1, a_2) \leftarrow$
$\quad \text{JOINTATTENTION}(B_1.\text{attn}, B_2.\text{attn}, \hat{x}_1, \hat{x}_2, \text{mask}_1, \text{mask}_2, \text{rope}_1, \text{rope}_2, \alpha_1, \alpha_2)$

/* 3. Residual connection with MSA gating */
$x_1 \leftarrow x_1 + g_1^{\text{msa}} \odot a_1$
$x_2 \leftarrow x_2 + g_2^{\text{msa}} \odot a_2$

/* 4. Feed-forward branch — stream 1 */
$\tilde{x}_1 \leftarrow B_1.\text{ff\_norm}(x_1) \odot (1 + e_1^{\text{ff}}) + s_1^{\text{ff}}$
$f_1 \leftarrow B_1.\text{ff}(\tilde{x}_1)$
$x_1 \leftarrow x_1 + g_1^{\text{ff}} \odot f_1$

/* 5. Feed-forward branch — stream 2 */
$\tilde{x}_2 \leftarrow B_2.\text{ff\_norm}(x_2) \odot (1 + e_2^{\text{ff}}) + s_2^{\text{ff}}$
$f_2 \leftarrow B_2.\text{ff}(\tilde{x}_2)$
$x_2 \leftarrow x_2 + g_2^{\text{ff}} \odot f_2$

**return** $(x_1, x_2)$

---

**Algorithm 2: JointAttention** – Multi-head attention with optional joint token pooling. Concatenation $[\,\cdot\,;\,\cdot\,]$ is along the sequence dimension. SDPA denotes scaled dot-product attention. CUSTOMDIAGMASK is a full+diagonal window mask, illustrated in Fig. 3.

---

**Input:** Attention modules $\texttt{attn}_1, \texttt{attn}_2$ with $QKV$ projections
Hidden states $x_1 \in \mathbb{R}^{B \times L_1 \times d}$, $x_2 \in \mathbb{R}^{B \times L_2 \times d}$
Optional rotary embeddings: $\texttt{rope}_1, \texttt{rope}_2$
Optional local window masks: $\texttt{mask}_1, \texttt{mask}_2$
Joint attention flags: $\alpha_1, \alpha_2 \in \{0, 1\}$
**Output:** Attention outputs $o_1 \in \mathbb{R}^{B \times L_1 \times d}$, $o_2 \in \mathbb{R}^{B \times L_2 \times d}$

---

**1. Project input to QKV:**
$(q_1, k_1, v_1) \leftarrow \texttt{attn}_1.\text{TOQKV}(x_1)$
$(q_2, k_2, v_2) \leftarrow \texttt{attn}_2.\text{TOQKV}(x_2)$

**2. Apply rotary embeddings (if provided):**
**if** $\texttt{rope}_1$ *exists* **then**
$\quad \llcorner\ (q_1, k_1) \leftarrow \text{APPLYROPE}(q_1, k_1, \texttt{rope}_1)$
**if** $\texttt{rope}_2$ *exists* **then**
$\quad \llcorner\ (q_2, k_2) \leftarrow \text{APPLYROPE}(q_2, k_2, \texttt{rope}_2)$

**3. Construct joint token pools:**
**if** $\alpha_1 = 1$ **then**
$\quad | \quad q_1^\star \leftarrow [q_1; q_2], \quad k_1^\star \leftarrow [k_1; k_2], \quad v_1^\star \leftarrow [v_1; v_2]$
**else**
$\quad \llcorner\ (q_1^\star, k_1^\star, v_1^\star) \leftarrow (q_1, k_1, v_1)$
**if** $\alpha_2 = 1$ **then**
$\quad | \quad q_2^\star \leftarrow [q_2; q_1], \quad k_2^\star \leftarrow [k_2; k_1], \quad v_2^\star \leftarrow [v_2; v_1]$
**else**
$\quad \llcorner\ (q_2^\star, k_2^\star, v_2^\star) \leftarrow (q_2, k_2, v_2)$

**4. Split heads and apply masks:**
$(q_1^\star, k_1^\star, v_1^\star) \leftarrow \text{SPLITHEADS}(q_1^\star, k_1^\star, v_1^\star)$
$(q_2^\star, k_2^\star, v_2^\star) \leftarrow \text{SPLITHEADS}(q_2^\star, k_2^\star, v_2^\star)$

**if** $\texttt{mask}_1 \neq \varnothing \wedge \alpha_1 = 1$ **then**
$\quad \llcorner\ M_1 \leftarrow \text{CUSTOMDIAGMASK}(L_1, L_2, \texttt{mask}_1)$
**else**
$\quad \llcorner\ M_1 \leftarrow \varnothing$
**if** $\texttt{mask}_2 \neq \varnothing \wedge \alpha_2 = 1$ **then**
$\quad \llcorner\ M_2 \leftarrow \text{CUSTOMDIAGMASK}(L_2, L_1, \texttt{mask}_2)$
**else**
$\quad \llcorner\ M_2 \leftarrow \varnothing$

**5. Compute scaled dot-product attention:**
$o_1^\star \leftarrow \text{SDPA}(q_1^\star, k_1^\star, v_1^\star, M_1)$
$o_2^\star \leftarrow \text{SDPA}(q_2^\star, k_2^\star, v_2^\star, M_2)$

**6. Merge heads, trim to original length, and project:**
$o_1 \leftarrow \text{MERGEHEADS}(o_1^\star)[:, : L_1], \quad o_1 \leftarrow \texttt{attn}_1.\text{OUTPROJ}(o_1)$
$o_2 \leftarrow \text{MERGEHEADS}(o_2^\star)[:, : L_2], \quad o_2 \leftarrow \texttt{attn}_2.\text{OUTPROJ}(o_2)$

**return** $(o_1, o_2)$

---

