# OpenReview forum: "JAM-Flow: Joint Audio-Motion Synthesis with Flow Matching"
_ICLR.cc/2026/Conference — ICLR 2026 Conference Desk Rejected Submission_

### Official Review · Reviewer_Qj8r · 2025-10-21

**Soundness:** 3
**Presentation:** 3
**Contribution:** 3
**Rating:** 4
**Confidence:** 4

**Summary:**

The paper propose JAM-flow, the first joint training framework for talking head and TTS generation. The proposed inpainting-style joint supervision effectively improves the performance.

**Strengths:**

1. The proposed method is novel. It is the first joint training framework for talking head and TTS generation.
2. The paper is well written and easy to follow.

**Weaknesses:**

1. The performance in demo video is not satisfactory. The teaser video exhibits significant artifacts.
2. The paper lacks comparisons with recent methods, such as EDTalk.
3. Since the paper only generate mouth motions, it should compare the method with visual dubbing methods (which also only generate mouth motions), such as wav2vec, stylesync.

**Questions:**

It would be appreciated if the authors can provide comparisons result with recent methods, such as EDTalk, etc.

typo: 2.2 TALKING HEAD HENERATION -> Generation.

---

> ### Author Response · Authors · 2025-11-25
>
> We thank the reviewer for recognizing the novelty of our joint training framework.
>
> **1. Teaser Video Artifacts**
> The teaser intentionally included out-of-domain (stylized/non-human) examples to demonstrate generalization, which naturally results in more artifacts than in-domain data. We also note that most artifacts come from the warping operation of the head poses which are derived from LivePortrait. Generated lip motions match the audio precisely. We will clarify this in the paper. We gently request the reviewer to look at the other supplementary videos, which showcase high-quality results on in-domain human inputs.
>
> **2. Additional Baselines**
> We appreciate the suggestion. We are adding comparisons with EDTalk and Wav2Vec-based dubbing methods to the supplementary material to provide a more comprehensive evaluation of mouth-motion generation. We’ll cite StyleSync as its model weights are currently unavailable.
>
> https://anonymous-submission12338nci.github.io/random-supplementary-web/
>
> **3. Typo**
> We have corrected "HENERATION" to "Generation" in Section 2.2. Thank you.
>
> Thank you once again. If you have any further questions, please do not hesitate to ask.

---

### Official Review · Reviewer_cug6 · 2025-10-31

**Soundness:** 3
**Presentation:** 3
**Contribution:** 2
**Rating:** 4
**Confidence:** 2

**Summary:**

This paper introduces a joint audio–motion flow matching framework that learns both modalities within a single model.
It combines Motion-DiT and Audio-DiT through joint attention and asymmetric temporal masking, aiming to improve synchronization between speech and facial motion.

**Strengths:**

1. The paper is clear and easy to follow, with well-structured writing and visuals.
2. It cites relevant work and explains design choices logically, making the reasoning convincing.
3. The flexible input setting is an interesting and practical aspect of the framework.

**Weaknesses:**

1. The paper does not provide clear quantitative evidence on how joint training improves performance over unimodal setups. Based on the current presentation, the main benefit of joint supervision seems to lie in enabling flexible input configurations rather than delivering measurable quality gains.
2. The effectiveness of the proposed attention masking is discussed qualitatively, but additional quantitative analysis would help clarify its impact on temporal alignment and overall performance.
3. The overall novelty feels incremental, as most components (F5-TTS, LivePortrait, MM-DiT) are reused.
Still, positioning it as the first joint, inpainting-style framework for flexible bidirectional audio–motion generation is reasonable, and specifying what new capabilities or effects arise directly from the joint setup would make the contribution clearer.

**Questions:**

1. Could the authors provide more details about the unimodal baselines—for example, whether the audio-only and motion-only branches were trained independently—and include both quantitative results and qualitative examples comparing these baselines to the joint model?
2. The paper discusses an asymmetric attention design. It would be helpful to include quantitative evidence showing how this design choice affects temporal alignment or overall performance.

I would be inclined to raise my rating if the authors address the questions and clarifications mentioned above.

---

> ### Author Response · Authors · 2025-11-25
>
> We appreciate the positive remarks on the paper's clarity and logical design.
>
> **1. Evidence for Joint Training**
> Our claim is that inpainting-based training effectively learns the joint distribution. While TTS WER is bounded by pseudo-labels, JAM-Flow achieves SOTA or competitive performance on THG and Dubbing against specialized models. To provide clearer evidence, we have added direct comparisons with cascaded pipelines to the supplement, showing that joint training yields superior synchronization and coherence. anonymized project page: https://anonymous-submission12338nci.github.io/random-supplementary-web/
>
> **2. Asymmetric Attention Masking**
> We provided quantitative evidence in Table 4 of the main paper. To recap:
> * **Without Mask:** FID 5.74, LSE-D 7.99
> * **With Mask:** **FID 5.66**, **LSE-D 7.73**
> The masking strategy consistently improves FID, LSE-D, and Semantic Similarity. We have added qualitative visualizations to the supplement to make the stabilizing effect of this design more tangible.
>
> **3. Novelty**
> We emphasize that our contribution is not replacing components like F5-TTS or LivePortrait, but integrating them into a unified flow-matching framework. This "pretrained-chaining + joint finetuning" approach offers:
> * **Rapid Convergence:** Stable synchronization emerges in ~10 minutes (400 steps).
> * **Unified Capability:** Solving multiple tasks with one model, which independently trained components cannot do.
> We will revise the introduction to better highlight this integration and training paradigm as the core novelty.
>
> **4. More information on the unimodal baseline**
> We appreciate the insightful question. As mentioned in line L309 of the paper, we first train the Stage-1 model. In this stage, the audio-only and motion-only branches are trained independently, as the reviewer mentioned.
> It's corresponding performance metric is briefly reported in Table 1. While both FID and FVD are strong, the lip-sync performance is slightly weaker. However, once we proceed to joint training, we observe improvements across all metrics.
>
> We will add more detailed explanations in the appendix.
>
> **5. Asymmetric attention design.**
> | # Joint Blocks | Attn. Mask |   FID ↓   | LSE-C ↑ | LSE-D ↓ |  WER ↓  | SIM-o ↑ |
> |----------------|---------|-----------|---------|---------|---------|---------|
> | 11 (Half)      | X         |   5.748   |  6.44   |  7.99   |  7.93%  |  0.61   |
> | 11 (Half)      | ✓       |   5.662   |  6.45   |  7.73   |  7.28%  |  0.64   |
>
> As shown in the table, applying the attention mask improves lip-sync performance. Additionally, both WER and FID also improve, indicating that the masking strategy provides performance gains across all tasks.
> We have included this ablation on the anonymized project page (attention-masking ablation tab):
> https://anonymous-submission12338nci.github.io/random-supplementary-web/

---

### Official Review · Reviewer_Zo8R · 2025-11-01

**Soundness:** 3
**Presentation:** 3
**Contribution:** 3
**Rating:** 4
**Confidence:** 3

**Summary:**

This paper proposes JAM-Flow, a unified framework for joint audio-visual synthesis that simultaneously generates speech and facial motion using flow matching. The method introduces two modality-specific diffusion transformers, Motion-DiT for implicit facial keypoint dynamics and Audio-DiT for speech generation — coupled through selective joint attention layers. Several architectural design choices, such as temporally aligned rotary positional embeddings (RoPE) and localized asymmetric attention masking, are proposed to enable stable and temporally consistent multimodal learning. Experimental results show improvements in synchronization and realism across multiple metrics.

**Strengths:**

*  joint modeling direction: The idea of training a single flow-matching framework to co-model both speech and facial motion is conceptually appealing and targets an underexplored space between talking-head generation and TTS systems.
* Architectural clarity: The paper carefully describes how Motion-DiT and Audio-DiT interact through partial joint attention, cross-modal RoPE alignment, and modality-specific attention masking, all of which are reasonable and technically sound design choices.
* Stability insights: The introduction of asymmetric masking and RoPE alignment are practical contributions that could generalize to other multimodal DiT architectures.
* Empirical completeness: The paper provides both quantitative results and ablation analyses, showing that the proposed fusion and masking yield modest but consistent performance improvements.

**Weaknesses:**

1. Partial joint attention design not well justified: Only half of the layers are fused via joint attention, but the paper does not explain how this number was selected, whether more or fewer fusion layers were tested, or how fusion depth affects stability and performance.
2. Unclear practical benefits of joint modeling: While the paper argues that jointly modeling speech and motion reflects the natural coupling of human communication, it is unclear what measurable benefit this joint training provides compared to separately trained TTS and talking-head models. The results do not convincingly demonstrate improvements in either lip synchronization, intelligibility, or overall audiovisual coherence due to the joint setup.
3. Unclear loss balancing across modalities: The final loss combines motion and audio velocity objectives, but no details are given on how they are balanced. Since these losses have different magnitudes and noise characteristics, simple summation is unlikely to be optimal. Techniques such as dynamic weight averaging or uncertainty weighting could improve stability. Without this clarification, reproducibility and convergence behavior remain ambiguous.
4. Weak and incomplete ablation study (Table 4):
The ablation does not cleanly isolate variables model depth, attention masking, and Audio-DiT finetuning are all changed simultaneously. The absence of a “Full + Mask” baseline makes it difficult to attribute improvements. The gains (e.g., FID 5.75 → 5.66, SIM 0.63 → 0.64) are marginal and lack variance reporting.
5. Inpainting-style supervision not novel: The “inpainting-style” joint supervision resembles existing masked-prediction training used in F5-TTS and other multimodal diffusion models, and its distinct contribution here is not experimentally validated.

**Questions:**

1.	What concrete advantages does joint modeling provide over independently trained TTS and talking-head models? Please quantify or illustrate how co-training improves synchronization, expressiveness, or audiovisual coherence. What are the scenarios you see this improvement clearly.
2.	How are the audio and motion velocity losses balanced during training? Given their different scales and variances, did you normalize or dynamically weight them to avoid one modality dominating the optimization?
3.	Why are only half of the Transformer layers fused through joint attention, and how sensitive are results to this choice? Was this proportion empirically tuned, or based on observed stability/performance trade-offs?
4.	In Table 4, the ablation mixes multiple design changes (depth, masking, Audio-DiT finetuning).
Could you provide a controlled ablation isolating each factor. For example, a “Full + Mask” configuration, to clarify where the gains come from?

---

> ### Author Response · Authors · 2025-11-25
>
> We thank the reviewer for the detailed feedback and for recognizing the clarity of our architecture and the stability provided by our masking and RoPE designs. For certain questions, we kindly ask reviewers to refer to the global response.
>
> **1. Partial (Half-Layer) Joint Attention**
> Our choice of fusing only the first half of the layers is a deliberate efficiency/performance trade-off.
> * It is empirically established (in models like Flux/Pyramid Flow) that early layers handle multimodal integration while later layers specialize.
> * Implementing the Full-joint attention with our asymmetric masking requires dual attention passes, which exceeds the memory limits of our 48GB VRAM setup.
> * Empirically (L420), we observed negligible performance difference between full and half fusion (especially during qualitative inspection), making half fusion the optimal design choice.
>
> **2. Loss Balancing**
> We normalize the token representations of each branch to have mean ($\approx 0$) and variance ($\approx 1$). Because the inputs are normalized, a simple 1:1 weighting of the velocity losses is effective. We found no meaningful improvement from dynamic weighting in our experiments.
>
> **3. Novelty of Inpainting-Style Supervision**
> While inpainting is used in fields like TTS (F5-TTS), our contribution is extending it to cross-modal joint supervision. This enables:
> 1.  Learning cross-modal dependencies without explicit alignment losses.
> 2.  Flexible, bidirectional conditioning (e.g., motion-to-audio).
> 3.  Solving three distinct tasks with one unified model.
> The empirical success across these tasks validates the novelty and effectiveness of this formulation.
>
> **4. Practical Benefits**
> The practical advantage is concrete: flexibility and coherence. A single JAM-Flow model handles TTS, talking-head, and dubbing. Furthermore, as shown in our Exclusive Use Cases (Supplement), joint modeling handles partial inputs (e.g., generating audio and motion simultaneously from text) with a naturalness that cascaded models fail to achieve.
>
> **5. Ablation Limitations**
> We agree that a Full + Mask configuration would additionally strengthen the analysis. However, this exceeds our current hardware constraints (48GB VRAM). We emphasize that the existing ablation study successfully isolates the benefits of both asymmetric masking and joint fusion, showing consistent improvements.
>
> Furthermore, we have included this ablation on the anonymized project page: https://anonymous-submission12338nci.github.io/random-supplementary-web/ (attention-masking ablation tab) to provide qualitative evidence. The results clearly show that without masking, the model fails to produce proper lip-synchronization.

---

### Official Review · Reviewer_pYmU · 2025-11-02

**Soundness:** 3
**Presentation:** 2
**Contribution:** 2
**Rating:** 4
**Confidence:** 4

**Summary:**

This paper presents a unified framework that integrates multiple tasks using Flow Matching and MM-DiT, along with an inpainting-style joint supervision strategy to support training across the TTS-to-THD pipeline. While the framework design and experimental coverage are reasonably thorough, the underlying motivation remains unconvincing. The method appears to be driven more by the absence of existing joint models than by a clearly defined problem or application need. Consequently, the contribution reads more as an engineering-level assembly of components than a conceptually motivated advancement.

**Strengths:**

1. Consolidating multiple functionalities into a unified model enhances flexibility and improves usability for uses.

2. The authors have made commendable efforts in both model functionality and experimental design, which may offer certain insights for future research.

3. The analysis of mouth keypoints in LivePortrait contributes to improved lip-speech consistency.

**Weaknesses:**

1. The authors claim to propose the first unified framework for TTS and THG, yet the motivation and significance of such integration are not clearly articulated. In typical THG pipelines, the driving signals—whether speech, video, or text—are already well-supported by a range of mature generation techniques, which raises questions about the practical necessity of unifying these components.

2. The combination of TTS and THG in this work appears more akin to an engineering optimization effort than a conceptually novel contribution. In practice, many developers have already achieved impressive results by chaining together multiple existing technologies. For example, in the creation of virtual idols.

3. The supplementary video reveals a noticeable issue in image-to-video generation: the background tends to shift along with the subject’s motion. This artifact is non-trivial and may affect perceived realism, yet the authors do not provide analysis or discussion regarding its cause or impact.

4. Weak TTS performance may negatively affect the accuracy of mouth movements, potentially lowering the overall effectiveness of the model.

5. Wav2Lip is a talking face generation method, yet it is incorrectly categorized as a TTS system in the first paragraph of the introduction.

**Questions:**

1. The output of facial expressions relies on mouth keypoints from LivePortrait, which raises the question of how well other expressions such as smiling and eye gaze are represented.

2. How long of a video does the model support generating?

3. The talking head experiments should include comparisons with more representative baselines such as Hallo2, Wav2Lip, and MEMO.

4. I would like to understand the performance gap when using a cascaded model to achieve the same goal. For example, in a text-audio-visual pipeline, the system first generates speech using TTS and then drives facial expressions. How does the performance of this approach compare to the method proposed in the paper?

---

> ### Author Response · Authors · 2025-11-25
>
> We thank the reviewer for acknowledging the flexibility and experimental breadth of our work. For certain questions, we kindly ask reviewers to refer to the global response.
> We address the concerns regarding motivation and artifacts below.
>
> **1. Motivation and Contribution**
>
> Our central claim (L93, L263, L457) is that joint generation via an inpainting objective allows the model to effectively learn the relationship between semantically related modalities. We believe that it is not merely an engineering combination of TTS and THG, but a conceptual contribution: a training paradigm that implicitly learns cross-modal consistency and enables flexible, bidirectional conditioning that cascaded pipelines fail to achieve. We will refine the text to emphasize this conceptual value.
>
> **2. Background Shift Artifacts**
>
> We acknowledge that our dependency on LivePortrait for efficient (~11 FPS) rendering inherits its known limitation regarding background drift during large motions. This is a trade-off for efficiency and representation compactness. We will explicitly discuss this as a limitation and an avenue for future work.
>
> **3. Baselines and Citations**
>
> * **Wav2Lip [1]:** We have corrected the citation error (L034). Regarding performance, Wav2Lip is trained directly on LSE metrics, making its high scores unreliable for fair comparison, a known issue noted in works like PortraitTalk [2]. Nevertheless, we will still include the Wav2Lip scores in our paper for completeness.
> * **Hallo2/3:** We compared against Hallo and Hallo3. We excluded Hallo2 [3] as it focuses on long-duration synthesis and shares authorship with Hallo3.
>
> **4. Expressiveness Beyond Mouth**
>
> Our design explicitly models the strong correlation between audio and mouth keypoints. For other expressions (e.g., eye-gaze/-smiles), we currently leverage appearance cues from the reference video. While we agree that prosody correlates with upper-facial expressions, we focused on the most dominant coupling (lip-sync) for this work as it is most tightly related to the audio signal.
>
> **5. Cascaded Pipeline Comparison**
>
> We have conducted the requested comparison between a cascaded pipeline ($\text{text} \rightarrow \text{audio} \rightarrow \text{facial motion}$) and our joint model. These results, added to the supplementary material, demonstrate the superior coherence of our joint approach. We add it on the anonymized project page: https://anonymous-submission12338nci.github.io/random-supplementary-web/ (cascaded vs. joint comparison tab)
>
> Overall, the joint model yields results that are significantly better synchronized compared to those produced by the cascaded pipeline.
>
> Reviewer may check additional results suggested by other reviewers, such as the attention-masking ablation tab.
>
> [1] Prajwal et al., “Wav2Lip: Accurately Lip-syncing Videos In The Wild”, ACM MM 2020
>
> [2] Nazarieh et al., “PortraitTalk: Towards Customizable One-Shot Audio-to-Talking Face Generation”, arXiv 2024
>
> [3] Cui et al., “Hallo2: Long-Duration and High-Resolution Audio-Driven Portrait Image Animation”, ICLR 2025

---

### Official Review · Reviewer_rF3g · 2025-11-08

**Soundness:** 3
**Presentation:** 3
**Contribution:** 2
**Rating:** 4
**Confidence:** 3

**Summary:**

This paper proposes a joint training framework for talking head and TTS generation. Specifically, it combines Motion-DiT and Audio-DiT modules through partial joint attention, RoPE alignment and asymmetric masking. The model handles multiple tasks—talking head generation, TTS, and video dubbing—within one architecture.

**Strengths:**

1. The idea to unify speech and motion synthesis in a single model is interesting. And the architecture design including joint attention and masking is reasonable.
2. Proposed inpainting-based training enables flexible conditioning under missing modalities.

**Weaknesses:**

1. In the experiment, the improvements over baselines are marginal or inconsistent. In TTS, WER even increases compared with baselines.
2. The scalability and generalization of proposed framework are not convincingly discussed. How the method performs with longer sequences, different speakers or higher-resolution videos?
3. The evaluation is limited. Metrics such as LSE-C/LSE-D are known to be unstable and no strong multimodal baselines are provided for fair comparison.

**Questions:**

1. Could the framework generalize beyond face–speech to other modality pairs (for example, gesture–audio)?
2. How does computational cost compare to independent training of the two unimodal models?

---

> ### Author Response · Authors · 2025-11-25
>
> We thank the reviewer for finding our unified model design and inpainting-based training strategy interesting and reasonable. We appreciate the recognition of our framework's flexibility.
>
> **1. Comparison with Baselines**
>
> The baselines used were task-specific models trained exclusively for single objectives. The fact that JAM-Flow achieves competitive results across all tasks using a single joint objective strongly supports the validity of our unified design. We will add a discussion highlighting this distinction in the revision. Please refer to global response for detail of WER score.
>
> **2. Scalability and Generalization**
>
> We appreciate these suggestions and clarify as follows:
>
> * **Longer Sequences:** Our inpainting-based training naturally supports sliding-window inference. By conditioning new segments on the tail of previous generations, we can generate arbitrarily long sequences without discontinuity. Note that generated quality is optimal around ~20 seconds (matching our training distribution), while 60s+ generation in a single pass may degrade alignment.
>
> * **Different Speakers:** The model generalizes robustly across speakers. The inpainting objective forces the model to learn cross-modal relationships independent of specific identities.
>
> * **Higher Resolution:** JAM-Flow natively supports 512px video. By utilizing the post-processing pipeline from LivePortrait, we can achieve stable 1024px outputs. This resolution is at the frontier of current academic talking-head models.
>
> * **Generalization to Other Modalities:** The proposed training paradigm is scalable to any pair of correlated modalities (e.g., audio-gesture). The core design principle (predicting masked regions using aligned context from another modality) implicitly enforces joint structure learning regardless of the specific domain.
>
> **3. Evaluation Metrics (LSE-C/D)**
> We acknowledge the known instability of LSE metrics. However, they remain standard in the field. To supplement these metrics, we also provide User Study (Appendix B.3) results which shows JAM-Flow’s robust performance: JAM-Flow achieved the best average ranking (1.29) against baselines and a 62.6% preference rate in automated video dubbing.
> We also add additional baselines, such as EDTalk, on the anonymized project page: https://anonymous-submission12338nci.github.io/random-supplementary-web/
> Reviewer may check additional results suggested by other reviewers, such as the cascaded vs. joint comparison tab or the attention-masking ablation tab.
>
> **4. Face-Speech Generalization**
> As noted above, the inpainting strategy is not specific to faces. Random multimodal masking naturally reveals cross-modal dependencies. For any modalities with strong temporal or semantic coupling, this method allows the model to learn the joint distribution effectively.
>
> **5. Computational Cost**
>
> While joint modeling increases VRAM usage by 1.5x, we observe extremely fast convergence. Surprisingly, the training roughly converges within 400 training steps (approx. 10 minutes on 4xA6000), achieving highly synchronized lip and audio. We attribute this to the favorable optimization loss landscape created by the inpainting objective, which encourages the model to efficiently capture cross-modal cues.

---

### Author Response · Authors · 2025-11-25
**Global Response2**

**On TTS Performance and WER**

We appreciate the reviewers’ thoughtful feedback concerning the reported WER scores. We’d like to clarify two key points: (1) the WER is bounded by our data quality, not our model capacity, and (2) this value does not reflect a meaningful degradation in perceptual quality.

**(1) Elevated WER originates from Whisper-generated pseudo-GT.**

As noted in Line 369 and Appendix B.2, JAM-Flow requires aligned video-audio-transcript triplets. Since no public dataset provides this, we use CelebV-Dub with transcripts generated by Whisper. Whisper is not error-free, and its transcription noise propagates into the pseudo-GT.
* Crucially, the LibriSpeech-PC [1]  paper reports 4.5% WER for Whisper-base.
* This is almost identical to the ~4.9% WER we observe. (L369)
Thus, our WER is bounded by the quality of the pseudo-labels, rather than representing a failure in modeling or optimization. We also note that dedicated TTS baselines utilize clean audio-only corpora or filter data purely for audio fidelity. In contrast, our requirement for paired facial motion forces us to rely on video-sourced data (CelebV-Dub) processed via source separation (Spleeter), which inevitably introduces artifacts and noise absent in standard TTS training.


**(2) ~5% WER is not indicative of degraded quality.**
Empirically, the perceptual TTS quality difference is negligible (please refer to supplementary audio). Multiple independent sources support that a WER in this range is not problematic:
* Microsoft’s Technical Report [2] positions 5–6% WER as human-level performance.
* Reports from IBM, Apple ML, Speechmatics [3,4,5] similarly describe $\le$ 5% WER as effectively acceptable, noting that improvements below this threshold offer diminishing practical returns.

Given this context, our ~4.9% WER is within the accepted "human-level" regime. As WER cannot fully capture multimodal capabilities, our qualitative results and complementary metrics demonstrate that the joint training effectively captures cross-modal relationships without sacrificing the audio quality.

[1] Meister et al., “LibriSpeech-PC: Benchmark for Evaluation of Punctuation and Capitalization Capabilities of end-to-end ASR Models”, arXiv 2023

[2] Xiong et al., “Achieving Human Parity in Conversational Speech Recognition”, IEEE/ACM TASLP 2017

[3] Saon et al., “English Conversational Telephone Speech Recognition by Humans and Machines”, Interspeech 2017

[4] Apple Machine Learning Research, “Humanizing Word Error Rate for ASR Transcript Readability and Accessibility”, Apple Machine Learning Research Blog, 2024

[5] Hughes, “The Problem with Word Error Rate (WER)”, Speechmatics Blog 2023

---

### Author Response · Authors · 2025-11-25
**Global Response1**

We sincerely thank all reviewers for their assessments and constructive suggestions. Their feedback highlights important aspects of our work, and we appreciate the opportunity to clarify the motivation, contributions, and empirical findings of JAM-Flow.

**Positioning of JAM-Flow**

As recognized by reviewers [rF3g, pYmU, cug6], a key strength of JAM-Flow lies in its inpainting-based multimodal training strategy, which enables flexible input configurations across speech, motion, and text. This unified training objective is central to our contribution: it allows a single model to handle three distinct tasks (TTS, talking-head generation, and video dubbing) within one architecture, without requiring separate task-specific models or additional finetuning.

We do not claim that JAM-Flow outperforms highly versatile and optimized large video models (e.g., Veo3), which typically aim to create general scene videos with audio from text conditions. Instead we target a smaller-scale, head-focused system with flexible input configurations, which is more common in academia. Compared to recent larger proprietary models, our approach runs much faster [~11 fps, in Appendix B.5] on a single consumer-grade GPU, offers more human-tailored features (reference audio, accurate lip motions, source detail preservation), and can be used to generate arbitrarily long videos. Compared to recent academic publications, our contributions can be summarized as follows:

* **Joint Modeling of Audio and Motion:** We propose the first framework to jointly model facial motion and speech. By learning the joint distribution rather than conditional chains, our model naturally develops shared cross-modal representations. This leads to emergent flexibility, allowing the system to handle diverse input combinations (e.g., text-only, audio-only, or mixed modalities). This unifies tasks previously required dedicated models and enables capabilities, such as video dubbing, that are infeasible with naive cascading of standard approaches.

* **Efficient Design:** We employ an inpainting-based flow-matching objective coupled with an optimized joint module design (e.g., asymmetric attention design and temporal fusion) to effectively handle distinct audio and motion domains with minimal overhead. This design yields remarkable efficiency in both training and inference: (1) fast and stable training convergence, yielding synchronized results in just ~400 steps, (2) low cost inference, allowing it to run on consumer GPUs, (3) and scalable generation for arbitrarily long videos via sliding-window inference and frame-by-frame decoding.

* **High Perceptual Quality:** JAM-Flow demonstrates competitive empirical performance across three distinct task benchmarks, producing highly natural and synchronized outputs within a single unified model. Given the known instability of automated metrics like LSE (noted by reviewer rF3g) and WER (discussed below), our performance is best evidenced by our User Study in Appendix B.3, which confirms high preference rates. We specifically highlight our model’s superior perceptual quality in lip-synchronization and audio-linked dynamics, the core generative components of our work. Crucially, we maintain competitive performance against specialized, computationally heavier baselines in standard tasks, while enabling novel generation scenarios where task-specific models cannot operate and no established benchmarks yet exist.

---

### Author Response · Authors · 2025-12-03
**Summary of Rebuttal and Key Clarifications (1/3)**

Dear Area Chair,

We understand the difficulties caused by the OpenReview incident and sincerely appreciate your efforts during this period. We had submitted a detailed rebuttal along with many new experimental results, but the sudden freeze stopped the process immediately. This was especially unfortunate for our submission, as we believe the discussion phase would have been important for addressing the reviewers’ concerns and potentially adjusting the borderline scores. It is regrettable that we could not fully communicate the updates we prepared.

We are confident that our response substantially addresses the main concerns about the quantitative metrics and the motivation for joint modeling. Below, we summarize the common issues raised by the reviewers and how our rebuttal and new experiments address them. We kindly ask you to consider this summary together with the detailed replies in each thread and our supplementary website:
https://anonymous-submission12338nci.github.io/random-supplementary-web

---
**0. Common Clarification: WER and TTS Performance**

Several reviewers raised concerns about Word Error Rate (WER) and its implications for TTS quality.
- Our model requires aligned video-audio-transcript triplets, which are not available in public form datasets. We therefore use CelebV-Dub with transcripts generated by Whisper.
- As reported in the paper and Appendix B.2, the LibriSpeech-PC benchmark shows 4.5% WER for Whisper-base, which closely matches the ~4.9% WER we obtain. In other words, our WER is effectively bounded by the noise in the pseudo ground-truth, rather than by a capacity or optimization issue of JAM-Flow.
- In contrast, dedicated TTS baselines are trained on clean audio-only data, whereas our requirement for paired facial motion forces us to rely on noisier, video-sourced data with source separation.

Multiple independent reports (e.g., from Microsoft, IBM, Apple ML, Speechmatics) characterize 5-6% WER as human-level, and further gains below this range have diminishing practical benefits. Our ~4.9% WER lies well within this human-level regime. Qualitatively, we observe negligible differences in perceptual TTS quality at this level, and our user study (Appendix B.3) corroborates that listeners still prefer JAM-Flow in dubbing scenarios.

Overall, we argue that WER in this range does not meaningfully harm perceptual quality and should be interpreted in the context of (i) noisy pseudo labels and (ii) the broader multimodal objectives of JAM-Flow, rather than as a failure of the model on specific task.

---
**1. Positioning and Novelty of JAM-Flow**

Positive assessment
- Reviewer rF3g finds “the idea to unify speech and motion synthesis in a single model” interesting and considers the joint attention and masking design reasonable.
- Reviewer pYmU notes that consolidating multiple functionalities into a unified model “enhances flexibility and improves usability” and acknowledges our efforts in model functionality and experimental design.
- Reviewer cug6 highlights that the paper is “clear and easy to follow” and finds the flexible input setting “interesting and practical.”
- Reviewer Qj8r describes our method as “the first joint training framework for talking head and TTS generation” and considers it novel and well written.

Concerns
- Reviewers pYmU and cug6 question whether the contribution is mainly an engineering combination of existing components (F5-TTS, LivePortrait, DiT), and whether the motivation for a joint framework is clearly justified beyond simply chaining existing systems.
- Reviewer Zo8R asks what concrete advantages joint modeling provides over independently trained TTS and talking-head models, especially in terms of measurable gains in synchronization and coherence.

Our response
- We clarified that JAM-Flow is positioned as a smaller-scale, head-focused system with flexible input configurations, which is more typical in academic settings, rather than as a competitor to larger and slower video models (e.g., Veo3).
- Our inpainting-based multimodal training strategy allows a single model to handle three tasks (TTS, talking-head, and video dubbing) within one architecture, without separate task-specific models or additional finetuning. This is enabled by learning a joint distribution rather than a conditional cascade, leading to shared cross-modal representations and emergent flexibility.
- We emphasized that this is not just an engineering combination but a training paradigm: joint inpainting-style supervision that implicitly learns cross-modal consistency and supports bidirectional conditioning, which cascaded pipelines cannot easily realize.
- We also highlighted scalability and practicality: JAM-Flow supports sliding-window inference for long sequences naturally because of joint inpainting-style training method, making it suitable for real-world scenarios while still being computationally efficient.

---

> ### Author Response · Authors · 2025-12-03
> **Summary of Rebuttal and Key Clarifications (2/3)**
>
> ---
>
> **2. Benefits of Joint Training vs. Unimodal and Cascaded Pipelines**
>
> Positive assessment
> - Reviewer Zo8R finds the joint modeling direction “conceptually appealing” and notes that our design targets an underexplored space between talking-head generation and TTS.
> - Reviewer cug6 considers our flexible input setting “interesting and practical” and notes that positioning the method as a joint, inpainting-style framework is reasonable.
> - Reviewer Qj8r states that the proposed joint supervision “effectively improves the performance.”
>
> Concerns
> - Several reviewers request clearer quantitative evidence that joint training outperforms unimodal or cascaded pipelines.
> - They question whether improvements go beyond flexibility.
>
> Our response
>
> We added several new experiments and clarifications:
> - Cascaded vs. Joint Pipeline:
>     - We implemented a cascaded baseline (Text→TTS→Talking Head) and compared it against JAM-Flow. The results, reported on the project page (“cascaded vs. joint” tab), show that our joint model produces noticeably better synchronization and coherence, especially in challenging dubbing scenarios. In particular, joint training improves naturalness and temporal alignment in ways that are not fully captured by existing metrics.
> - Unimodal Baseline (Stage-1):
>     - While Stage-1 achieves competitive FID/FVD, its lip-sync performance is weaker. Once we proceed to joint training, we observe consistent improvements across all metrics (including lip-sync), demonstrating that joint supervision adds value beyond flexibility.
> - User Study:
>     - Our user study (Appendix B.3) further supports these findings: JAM-Flow achieves the best average ranking (1.29) in talking-head and a 62.6% preference rate in automated video dubbing. This directly reflects perceived synchronization and overall coherence and complements the numerical metrics.
> - Practical Advantages:
>     - We emphasized that joint training also yields practical benefits:
>         - one unified model instead of multiple task-specific models, which can handle partial inputs (e.g., only text, only audio, or mixed).
>
> Overall, these results show that joint training provides both practical flexibility and measurable improvements over unimodal and cascaded setups.
>
> ---
>
> **3. Architectural Design: Partial Joint Attention, Inpainting Supervision, and Masking**
>
> Positive assessment
> - Reviewer Zo8R praises our architectural clarity, noting that the interaction between Motion-DiT and Audio-DiT via partial joint attention, RoPE alignment, and masking is “reasonable and technically sound.”
> - Reviewer rF3g finds the inpainting-based training strategy interesting and acknowledges its ability to support flexible conditioning.
> - Reviewer cug6 finds our design choices logical and convincing.
>
> Concerns
> - Reviewer Zo8R questions the rationale behind fusing only half of the Transformer layers, loss balancing, the novelty of our inpainting supervision, and whether ablations cleanly isolate each design component.
> - Reviewer cug6 requests additional analysis for the asymmetric attention design and more details on the unimodal baseline.
>
> Our response
> - Partial Joint Attention (Half Layers):
>     - We explained that fusing only the first half of the layers is an intentional efficiency–performance trade-off:
>         - Prior work (e.g., Flux, Pyramid Flow) suggests that earlier layers handle multimodal integration, while later layers specialize in modality-specific refinement.
>         - Empirically, we observed negligible differences, making half fusion the most practical choice given our computational constraints. (we also report quantitative results)
> - Loss Balancing:
>     - We normalize token representations of each branch to have zero mean and unit variance, which allows a simple 1:1 weighting of the audio and motion velocity losses. We did not observe meaningful gains from more complex dynamic weighting strategies, and the current setup remained stable and reproducible.
> - Inpainting-Style Cross-Modal Supervision:
>     - While inpainting has been used in unimodal settings (e.g., F5-TTS), our contribution is to extend it to cross-modal joint supervision, enabling:
>         - learning cross-modal dependencies without explicit alignment losses,
>         - flexible, bidirectional conditioning (e.g., motion-to-audio as well as audio-to-motion), and
>         - solving three tasks (TTS, talking-head generation, dubbing) within a single, unified objective.
> We further noted that this inpainting strategy is applicable to other modality pairs (e.g., audio–gesture), indicating that the idea is not tied to the specific modalities used in this work.
> - Asymmetric Masking and Ablations:
>     - We already provided quantitative evidence that asymmetric attention masking improves both FID and lip-sync metrics.
>     - On the project page (“attention-masking ablation” tab), we added visualizations which show that removing masking leads to noticeably worse lip-sync behavior.

---

> ### Author Response · Authors · 2025-12-03
> **Summary of Rebuttal and Key Clarifications (3/3)**
>
> ---
>
> **4. Visual Artifacts, Baselines, and Evaluation Completeness**
>
> Positive assessment
> - Reviewer pYmU acknowledges that our analysis of LivePortrait mouth keypoints contributes to improved lip-speech consistency.
> - Reviewer Qj8r recognizes the novelty of our joint framework and finds the paper well written and easy to follow.
>
> Concerns
> - Reviewer pYmU notes background drift artifacts in the supplementary video and raises concerns about their impact on realism.
> - Reviewer Qj8r finds the teaser video unsatisfactory due to artifacts and requests comparisons with recent methods such as EDTalk.
> - Few reviewers (e.g., rF3g, cug6) ask for more baselines and more representative comparison sets.
>
> Our response
> - Background and Teaser Artifacts:
> We clarified that the teaser intentionally includes out-of-domain, stylized or non-human examples to demonstrate generalization, which naturally yield more artifacts than in-domain human inputs. Most visible artifacts arise from the LivePortrait warping-based renderer, which trades background stability for efficiency and compact representation and is mainly trained on facial movements rather than full-scene consistency. We will explicitly state this as a limitation and outline improvements (e.g., more advanced renderers) as future work.
> - Additional Baselines:
>     - We expanded and clarified our baselines by adding:
>         - EDTalk and Wav2Vec-based dubbing methods in the supplementary material to better reflect recent progress in visual dubbing.
>         - Wav2Lip scores, despite its training directly on LSE metrics (which makes direct comparison less reliable, as noted in PortraitTalk), so that our evaluation remains as complete and transparent as possible.
>         - Clarifications regarding Hallo/Hallo2/Hallo3 and why Hallo2 (which focuses on long-duration scenarios) was not used as the main comparison despite overlapping authorship with Hallo3.
> - Evaluation Beyond Automated Metrics:
>     - Recognizing the limitations of LSE and WER, we supplemented them with:
>         - a user study demonstrating clear preference for JAM-Flow in dubbing scenarios, and
>         - additional qualitative examples and long-sequence demos, including exclusive use cases, on the project page.
>
> These additions strengthen our claim that JAM-Flow maintains competitive or better perceptual quality compared to specialized baselines, while enabling scenarios where task-specific models and existing benchmarks do not naturally apply.
>
> ---
> **Closing**
>
> In summary, we believe JAM-Flow offers a meaningful conceptual and practical contribution by jointly modeling audio and motion within a single, efficient framework that supports flexible conditioning and multiple tasks. Our rebuttal and additional experiments provide:
> - A clearer positioning of the method and its novelty relative to cascaded and unimodal approaches,
> - A detailed analysis of WER and metric limitations, showing that our performance is effectively human-level given the pseudo labels,
> - New comparisons with cascaded pipelines and more baselines (EDTalk, Wav2Lip), and
> - Expanded ablations and user studies in appendix demonstrating the benefits of joint training, masking, partial joint attention, and inpainting-style cross-modal supervision.
>
> We kindly ask you to take these clarifications and new results into account when assessing the paper, especially given that the OpenReview freeze prevented reviewers from fully engaging with our updates.
>
> Thank you very much for your time and effort.
>
> Best regards,
>
> The Authors

---

### Note · Program_Chairs · 2026-01-17
**Submission Desk Rejected by Program Chairs**

The following references in this submission do not refer to real documents and/or have major errors in bibliographic information:

 Junyang Chen, Chenpeng Du, Zhenhui Ye, and Yanwei Fu. F5-TTS: High-fidelity text-to-speech via conditional flow matching and inpainting. arXiv preprint arXiv:2411.00000, 2024.